# Optimization of Fermentation Conditions for Increasing Erucamide Content in *Bacillus megaterium* Using Several Accelerants

**DOI:** 10.3390/microorganisms13010108

**Published:** 2025-01-08

**Authors:** Hao Zhao, Yudan Xie, Zhu Li, Longfeng Wei, Renli Ai

**Affiliations:** 1Key Laboratory of Plant Resource Conservation and Germplasm Innovation in Mountainous Region (Ministry of Education), College of Life Sciences/Institute of Agro-Bioengineering, Guizhou University, Guiyang 550025, China; HZhaosd@163.com (H.Z.); xieyudan921@163.com (Y.X.); lfweigz@163.com (L.W.); arl6688@163.com (R.A.); 2Guizhou Key Laboratory of Agricultural Biotechnology, Guiyang 550009, China

**Keywords:** *Bacillus megaterium*, erucamide, accelerant, metabolic mechanism

## Abstract

As a food packaging sliding agent, erucamide is widely used in the field of food packaging, but the traditional synthesis method of erucamide faces the problems of insufficient raw materials and low yield of colza oil. Our laboratory has found that *Bacillus megaterium* L2 has the potential to produce erucamide. This study aims to improve the ability of *B. megaterium* L2 to produce erucamide by adding various accelerants to optimize the fermentation conditions. Univariate and orthogonal tests showed that 0.3% Tween 80, 0.004% Ca^2+^, 0.04% colza oil, and 0.02% chloroform were the best regulation conditions for erucamide production of the L2 strain, and erucamide content reached 1.778 mg/L, which was 32.59% higher than the blank group and 60.26% higher than before fermentation culture. The mechanism of membrane metabolism in the L2 strain was further investigated, and our data suggested that the conductivity, nucleic acid and protein content, and β-galactosidase activity of L2 were increased significantly after treatment with accelerants, indicating that the accelerants changed the cell membrane permeability of the L2 strain but did not harm or kill the bacteria. Moreover, GC-MS analysis of the cell membrane fatty acids of the L2 strain showed that the ratio of unsaturated to saturated fatty acid components increased from 0.893 to 1.856, which increased the fluidity and reduced the rigidity of the cell membrane. This study could provide some theoretical reference for microbial erucamide fermentation.

## 1. Introduction

Erucamide (13-cis-docosenamide, C_22_H_43_NO), originally known as plant lipid, is a compound with neuroactivity in humans and animals. It has the same or similar properties as other fatty acid amides, such as the endogenous cannabinoid analogue oleoyl ethanolamine (OEA) [1]. Erucamide is widely used in plastics, engineering plastics, cosmetics, environmental protection coatings, food packaging and other fields as an excellent anti-binder, dispersant, lubricant, film release agent, and antistatic agent. As a kind of food packaging, aluminum–plastic composite film is often used in food packaging with a long shelf life or high gas sensitivity. The combination of erucamide and oleamide as slippery agents can reduce the internal friction of food packaging [2,3]. In addition, erucamide also has certain biological activities. Studies have found that erucamide in radish leaves can inhibit acetylcholinesterase and thus effectively prevent memory defects [4]. In addition, mouse experiments have shown that erucamide can also play an effective anti-anxiety role [5]. Red yeast rice has many physiological functions such as lowering blood pressure, lowering blood sugar, antibacterial, anti-tumor, and lowering blood ammonia. Erucamide is isolated from its natural products [6], indicating that erucamide may be related to the physiological functions of red yeast rice. Therefore, erucamide has great application value in the field of medicine and food.

*Bacillus megaterium* belongs to the genus *Bacillus*, which is widely distributed in nature, such as in soil, seawater, sediment, paddy field, dry grain, honey, and milk [7]. As a kind of beneficial functional bacteria, *B. megaterium* has the advantages of environmental friendliness, low nutritional requirements, simple culture conditions, fast growth, is harmless to humans and animals, and has strong anti-stress properties [8], which has attracted widespread attention and in-depth research. *B. megaterium* is a rich source of secondary metabolites. Secondary metabolites have broad-spectrum antibacterial activities [9], which mainly inhibit the growth of pathogens by producing antagonistic metabolites during growth and development, and even kill pathogens. In addition, *B. megaterium* also has a strong secretory capacity and can produce many extracellular enzymes and other substances [10].

Currently, research on improving the production of microbial metabolites has become a hot topic. Fu et al. found that the yield of Tremella polysaccharide can be effectively increased by high-pressure assisted extraction and subcritical pressurized water extraction [11]. Jia et al. found that enhancing the hypnotic effect of edible fungi also lies in increasing the metabolic amount of certain polysaccharide products of edible fungi [12]. Zhang et al. discovered the important role of Ganoderma lucidum polysaccharide in the cosmetics field, and how to increase the production of Ganoderma lucidum polysaccharide has also become their research direction [13]. At present, the main way to obtain erucamide is through chemical synthesis. The synthesis methods include direct ammonolysis of high erucic acid colza oil, erucic acid ammonification, erucic acid methyl ester ammonification, erucic acid acyl chloride ammonification, erucic anhydride ammonification, erucamide replacement, erucic acid nitrile hydrolysis, and erucamide synthesis. Fatty acid amides are usually prepared by the reaction of fatty acids with anhydrous ammonia at 200 °C and 345~690 kPa [14]. Erucamide, as a fatty acid amide, can also be obtained by this method. Awasthi et al. explored the production process of erucamide. They used microwave radiation as a heating element to react erucic acid with gaseous ammonia released from urea in the presence of a catalyst (diammonium hydrogen phosphate) at atmospheric pressure and high temperature. The reaction time is very short and the yield can reach 92% [15]. Sethi et al. also developed a new method for the synthesis of fatty acid n-derivatives, starting from the readily available reagent acetonitrile, for the purification of the produced erucamide [16]. However, these methods still face the problems of insufficient erucic acid raw materials, complex preparation process, high energy consumption, low product purity, and low yield. In addition, there are few reports on the establishment of microbial erucamide determination methods and the production of erucamide by microbial fermentation. It has been reported that the use of surfactants and other accelerants can improve the ability of strains to produce metabolites. Meng et al. found that Tween 80 can promote the fermentation of *Streptococcus* to produce polysaccharides [17]. Wei et al. found that surfactants SDS and CTAB could promote the growth of *S. cerevisiae* and increase the production of extracellular glutathione [18].

On the other hand, as far as we know, the addition of accelerants can have certain effects on bacterial cell membranes. The cell membrane of a microorganism serves as a critical barrier, controlling the entry and exit of metabolites, ions, and other molecules. It is essential for maintaining cellular homeostasis, energy balance, and metabolism. Changes in membrane permeability can have profound effects on microbial metabolite production, as the movement of metabolites, substrates, and waste products is directly linked to membrane transport processes. Measuring β-galactosidase activity provides an indirect measure of membrane permeability. Under normal conditions, the enzyme is typically retained within the cell, where it catalyzes the hydrolysis of lactose. However, if the cell membrane is damaged, β-galactosidase or its substrates may leak into the surrounding medium. By measuring the level of β-galactosidase activity in the extracellular space, researchers can gain insight into the degree of membrane permeability and the overall integrity of the microbial membrane [19]. Altering membrane permeability can influence the availability of precursors, the accumulation of metabolites, and the efficiency of microbial metabolism, which in turn affects industrial bioprocesses, antibiotic resistance, and cell survival [20].

In previous studies, five bioactive compounds were isolated from *B. megaterium* L2 by bioactivity-guided separation, and the characterization of compounds was performed by NMR (^1^H-NMR and ^13^C-NMR) and was identified as erucamide, behenic acid, palmitic acid, phenylacetic acid, and β-sitosterol, and a method for the determination of erucamide was established [21,22]. However, the erucamide production in the L2 strain was low. In this study, single-factor experiments and response surface methodology (RSM) were used to design experiments [23]. Different accelerants were added to the fermentation medium to improve the efficiency of erucamide production by the L2 strain, and the effect of accelerants on the metabolic mechanism of L2 cells was preliminarily studied.

## 2. Materials and Methods

### 2.1. Strains, Media, and Culture Conditions

*B. megaterium* L2 was isolated, screened, and identified by the Institute of Fungal Resources, Guizhou University. It is now preserved in the China Center for Type Culture Collection (CCTCC) with the accession number CCTCC NO. M2012381. According to the previous optimization experiments in the laboratory, the optimized medium for *B. megaterium* L2 was obtained. The components were glucose 5 g/L, peptone 16.40 g/L, beef extract 3 g/L, sodium chloride 3 g/L, pH: 5.80, and 121 °C high temperature sterilization for 20 min [24]. The formula for the NA medium was peptone 10 g/L, beef extract 3 g/L, sodium chloride 5 g/L, and agar 15–20 g/L, and autoclaved at 121 °C for 20 min. The fermentation medium mentioned in this paper refers to the addition of the corresponding promoter on the basis of the optimized medium.

Preparation of seed liquid. The preserved strain was inoculated on an NA plate, and activated by an inverted culture at 37 °C for 24 h. The activated strain was transferred to the optimized medium, and cultured at 30 °C and 150 r/min for 48 h to obtain the fermented seed liquid.

Preparation of fermentation broth. The seed liquid was inoculated into the optimized medium according to a 2% (*v*/*v*) inoculation amount, and the liquid volume was 50 mL/100 mL. The fermentation broth was cultured at 30 °C and 150 r/min for 48 h, and the corresponding accelerant was added to obtain the fermentation broth.

### 2.2. Instruments and Equipment

Agilent gas–liquid phase mass spectrometer, Agilent, California, USA; bS110S Precision Electronic Balance Beijing Saiduoli Balance Co., Ltd., Beijing, China; sX2-4-10 mini-centrifuge Thermo Company, Waltham, Massachusetts, USA; eYELA N-1100 Rotary Evaporator Shanghai Allen Instrument Co., Ltd., Shanghai, China; dK-98-II electric thermostatic water bath Tianjin Taisite Instrument Co., Ltd., Tianjin, China; mLS-3750 independent high temperature and high pressure autoclave Beijing Nuohuicheng Technology Co., Ltd., Beijing, China; bIOMATE 3S UV-VIS Spectrophotometer Seymour Fisher Technology (China) Co., Ltd., Beijing, China; dDB303A Portable Conductivity Meter Shanghai LeiMagnetic Instrument Factory, Shanghai, China.

### 2.3. Screening and Optimization of Accelerants

#### 2.3.1. Effects of Different Accelerant Types on Erucamide Production

Different types of accelerants were added to the optimized fermentation medium, and the fermentation medium without accelerants was used as a control to compare the effects of different accelerants on the production of erucamide by *B. megaterium* L2 (Table 1).

#### 2.3.2. Effects of Different Accelerant Additions on Erucamide Production

On the basis of the optimal promoter type, different concentrations of accelerants were added to the optimized fermentation medium to further explore the optimal addition amount of the corresponding accelerants (Table 2).

#### 2.3.3. Effect of Different Accelerant Addition Time on Erucamide Yield

On the basis of the best accelerant type and addition amount, the corresponding accelerants were added to the optimized fermentation medium at different times. The accelerants selected were 0.2% Tween 80, 0.05% chloroform, 0.006% Ca^2+^, and 0.5% colza oil. The experimental time gradient was set to 0 h, 6 h, 12 h, 24 h, and 36 h. the optimal addition time of the corresponding accelerant was further explored.

#### 2.3.4. Optimization of Accelerant Addition Combination by Orthogonal Method

Orthogonal design was determined according to the method described by Yu et al. [25]. According to the results of single factor analysis of the type and amount of accelerants, Tween 80 (A), chloroform (B), colza oil (C), and Ca^2+^ (D) were selected as four factors to investigate three levels. The L_9_ (3^4^) orthogonal experiment was designed with the content of extracellular erucamide as the index to obtain the best combination of accelerants for the production of erucamide by *B. megaterium* L2 (Table 3). Tween 80 and chloroform were added before fermentation, and Ca^2+^ and colza oil were added at 6 h of fermentation.

#### 2.3.5. Assay Method

The bacterial growth was determined by the dilution coating method. The fermentation broth was diluted with sterile water to different gradients. The diluted sample (1 mL) was added to the NA medium and uniformly coated to dry. The culture was inverted at 37 °C for 24 h. The number of viable bacteria in the bacterial solution was determined by the plate colony counting method.

The fermentation broth was centrifuged at 10,000 r/min for 10 min, and the supernatant was extracted with dichloromethane at 1:1 (*v*/*v*) for 2 h. The organic phase (lower layer) was concentrated and dried by rotary evaporator at 25~30 °C, and then dissolved with 2 mL dichloromethane. The appropriate solution was filtered with 0.22 μm microporous membrane and tested.

GC-MS chromatographic conditions were HP-5 chromatographic column (Agilent, 30 m × 0.25 mm × 0.25 μm) and inlet temperature 290 °C. The initial column temperature was 150 °C, maintained for 2 min, and increased to 300 °C at 20 °C/min for 3 min. The carrier gas was He, the flow rate was 1.0 mL/min, and the injection volume was 1 μL.

### 2.4. Effects of Accelerants on the Metabolism of B. megaterium L2 Cells

#### 2.4.1. Effects of Accelerants on the Conductivity of *B. megaterium* L2 Cells

The conductivity was determined according to the method of Lee et al. [26]. The L2 strain was inoculated into the fermentation medium for culture, and the accelerant was added according to the optimized type and parameters. The culture without any accelerant treatment was used as the control group. After 0, 2, 4, 6, 8, 10, and 12 h, 5 mL of culture was centrifuged at 10,000 r/min for 5 min, and the supernatant was taken. The conductivity of the bacterial suspension of the L2 strain was directly measured by a conductivity meter. Each treatment was repeated three times to observe the change in the conductivity of the *B. megaterium* solution with the treatment time.

#### 2.4.2. Effects of Accelerants on Nucleic Acids and Extracellular Proteins of *B. megaterium* L2 Cells

The L2 strain was inoculated into the fermentation medium for culture. Accelerants were added according to the optimized types and parameters, and the culture without any accelerant treatment was used as the control group. After 0, 2, 4, 6, 8, 10, and 12 h of treatment, 5 mL of culture medium was centrifuged at 10,000 r/min for 5 min, the supernatant was filtered by a 0.22 μm filter membrane, the absorbance at 260 nm and 280 nm in the supernatant was measured, and the changes in nucleic acid and protein with the treatment time were recorded. Each test was repeated three times.

#### 2.4.3. Effects of Accelerants on Extracellular β-GALACTOSIDASE Activity of *B. megaterium* L2

In our study, using o-nitrophenyl-β-D-galactoside (oNPG) as a substrate to determine the activity of β-galactosidase in the *B. megaterium* L2 bacterial suspension, the permeability of the cell membrane could be explored. The activity of β-galactosidase was determined according to the method of Liao et al. [27] with some modifications. The L2 strain was inoculated into the fermentation medium for culture, and the accelerant was added according to the optimized type and parameters. The culture without adding any accelerant was used as the control group. After 0, 2, 4, 6, 8, 10, and 12 h of treatment, 5 mL of L2 strain cell suspension was centrifuged at 10,000 r/min for 10 min, and the supernatant was taken as the enzyme solution for determination. An amount of 2.5 mL phosphate buffer (pH: 7) with 0.1 mol/L was added to the clean plugged test tube, and the test tube was incubated at 37 °C for 3 min. Then, 100 μL oNPG solutions with 4 mg/mL and 100 μL enzyme solution were added. A 100 μL 0.1 mol/L phosphate buffer instead of enzyme solution was used as a control. The reaction was carried out in a 37 °C constant temperature water bath for 10 min, and then 0.3 mL 1 mol/L Na_2_CO_3_ solution was added to terminate the reaction. After standing for 5 min, the absorbance was measured at 420 nm.

#### 2.4.4. Analysis of Fatty Acid Composition of *B. megaterium* L2 Cell Membrane After Treatment with Accelerant

The determination method was slightly modified according to the method of Ahn et al. [28]. The L2 strain was inoculated into the fermentation medium for culture, and the accelerant was added according to the optimized type and parameters. The culture without adding any accelerant was used as the control group. The fermentation broth was centrifuged at 10,000 r/min for 10 min to collect bacteria, and 0.04 g of bacteria was weighed and placed in a 50 mL centrifuge tube. First, 3 mL of 0.5 mol/L potassium hydroxide/methanol (7:250, *m*/*v*) solution was added, esterified in a 60 °C water bath for 15 min, and cooled to room temperature. Then, 3 mL of 5% sulfuric acid/methanol (1:19, *v*/*v*) solution was added, esterified in a 60 °C water bath for 15 min, cooled in an ice bath, and finally n-hexane was added at a constant volume to 10 mL. After shaking, it was allowed to stand until the solution was layered, and the supernatant was drawn into another test tube. The vortex mixer was used to mix it evenly. The 0.22 μm filter membrane was filtered and tested.

GC conditions were Agilent 6850, hP-5 chromatographic column (30 m × 0.25 mm × 0.25 μm), and inlet temperature 290 °C. The initial temperature of the chromatographic column was 180 °C, maintained for 0.5 min, heated to 215 °C at 6 °C/min, then heated to 230 °C at 3 °C/min, and maintained for 10 min. The carrier gas was He, the flow rate was 1.0 mL/min, and the injection volume was 1 μL.

MS conditions were ionization mode EI, electron energy 70 ev, ion source temperature 200 °C, transmission line temperature 200 °C, scanning range 35~450 amu, and full scanning mode. The solvent was delayed for 15 min and run for 80 min.

### 2.5. Statistical Analysis

All experimental data in this experiment were statistically analyzed and plotted using Excel 2016, Graphpad prsim 5, and Origin 2019, and response surface analysis was performed using Design Expert 8.06 software. The significance analysis was performed using IBM SPSS Statistics 19.0, and the significance test was performed using the “Duncan” method. The different lowercase letters in the data graph of the paper indicate that the difference was significant (*p* < 0.05). All experiments were three replicates.

## 3. Results

### 3.1. Effect of Accelerants on Erucamide Production

#### 3.1.1. Effects of Accelerant Types on Erucamide Production

The 0.1% (g/L) TritonX-100, SDS, and CTAB had significant inhibitory effects on the growth of the L2 strain and the production of erucamide (Figure 1a). Tween 80 and PEG-4000 promoted the growth of the strain and increased the yield of erucamide. Among them, Tween 80 had the most significant promotion effect (*p* < 0.05), reaching 1.634 ± 0.030 mg/L. Therefore, the addition amount of Tween 80 was selected for further screening.

As shown in Figure 1b, Zn^2+^ had a significant inhibitory effect on cell growth and erucamide production, indicating Zn^2+^ was not conducive to the growth of the strain and hindered the synthesis of erucamide. However, Mg^2+^, Ca^2+^, and Mn^2+^ showed a significant promotion effect compared with the blank group (*p* < 0.05). The results showed that the three metal ions promoted the increase in erucamide production, among which Ca^2+^ had the best effect. Subsequently, the effect of Ca^2+^ metal ion addition on cell growth and erucamide production was selected to determine the optimal metal ion and its addition concentration.

Organic solvents have been proved to be an additive that can effectively increase the amount of secondary metabolites and mycelia of microorganisms. For example, chloroform, ethanol, toluene, acetone, and other organic solvents have a certain promoting effect on the secondary metabolites of different strains [29]. The effect of organic solvents on the growth of the L2 strain was investigated by measuring the OD600 value of the fermentation broth with different solvents. Amounts of 0.1% (*v*/*v*) of n-butanol, methanol, absolute ethanol, chloroform, dichloromethane, n-hexane, and DMSO were added to the fermentation medium, and the culture without any organic solvent was used as the blank control (CK; in the following text, CK has the same meaning). After adding chloroform, the yield of erucamide increased the most obviously, reaching 1.529 ± 0.007 mg/L, which was 14.76% higher than that of the blank group (Figure 1c). N-butanol, dichloromethane, and n-hexane had little effect on the production of erucamide, but, after adding 0.1% DMSO, the yield of erucamide decreased significantly (*p* < 0.05). It was speculated that the effect of organic solvents on the growth of the L2 strain and erucamide was related to the nature of the organic solvents themselves. Because chloroform in the organic solvents to be tested had the most obvious increase in the yield of erucamide, the addition amount and addition time of chloroform were further screened.

#### 3.1.2. Effect of Accelerant Concentrations on Erucamide Yield

It can be seen that, with the increase in Tween 80 concentration, the growth of the L2 strain and the yield of erucamide also increased (Figure 2a). When the concentration was 0.2% and 0.3%, there was no significant difference in the yield of erucamide, which was 1.713 ± 0.005 mg/L and 1.705 ± 0.023 mg/L, respectively, which was 26.82% and 26.02% higher than the control. The above results showed that Tween 80 had no significant effect on the growth of bacteria and promoted the production of erucamide. Among them, 0.2% Tween 80 promoted the effect more significantly. Therefore, 0.2% Tween 80 was selected as the best surfactant to be added to the fermentation medium.

The addition of different amounts of chloroform had a significant effect on erucamide fermentation (Figure 2b). With the increase in chloroform concentration, the growth of the strain and the content of erucamide showed a trend of increasing first and then decreasing. The concentration of 0.01~0.5% chloroform significantly promoted the synthesis of erucamide in the L2 strain (*p* < 0.05), and the yield of erucamide was similar at 0.05% and 0.1% concentrations. When the concentration of chloroform was more than 0.1%, the ability of the strain to synthesize erucamide began to decrease. Therefore, 0.05% was selected as the optimal concentration of chloroform. At this time, the content of erucamide was 1.544 ± 0.015 mg/L, which was 16.02% higher than that of the blank control.

Compared with the blank group, different amounts of Ca^2+^ promoted the production of erucamide, and, with the increase in concentration, the ability of the L2 strain to synthesize erucamide showed low promotion and high inhibition (Figure 2c). Compared with the blank group, 0.006% Ca^2+^ had a significant effect on the growth of L2 cells and the increase in erucamide production. Therefore, the best metal ion selection was 0.006% Ca^2+^.

With the increase in colza oil concentration, the growth and erucamide content of the L2 strain showed different degrees of increase (Figure 2d). Among them, 0.1~1% colza oil significantly promoted the growth of the L2 strain (*p* < 0.05), and the erucamide production at three concentrations was 1.664 ± 0.012 mg/L, 1.695 ± 0.005 mg/L, and 1.641 ± 0.020 mg/L, respectively, which was 23.66%, 25.94%, and 21.99% higher than that of the blank control. Therefore, the considered selection was a 0.5% added concentration of colza oil.

#### 3.1.3. Effect of Accelerant Addition Time on Erucamide Production

Concentrations of 0.2% Tween 80, 0.05% chloroform, 0.006% Ca^2+^, and 0.5% colza oil were added to the fermentation medium at different fermentation times (0 h, 6 h, 12 h, 24 h, and 36 h, respectively). The effect of the addition time of the four accelerants on the growth and erucamide content of the L2 strain was observed with the optimized medium. The addition of Tween 80 at different times had a significant effect on the synthesis of erucamide (*p* < 0.05) (Figure 3a). The addition of Tween 80 at 0~36 h promoted the increase in erucamide content first and then inhibited it. Therefore, 0.2% Tween 80 was added at 0 h, and the yield was 1.714 ± 0.024 mg/L, which was 29.78% higher than that of the blank control. The results of the effect of the chloroform addition time on erucamide production showed that the addition of chloroform at different fermentation times significantly promoted the synthesis of erucamide (*p* < 0.05) (Figure 3b) but the yield of erucamide decreased with the addition time of chloroform. In summary, the optimal addition time of 0.01% chloroform before fermentation (0 h) was selected. Moreover, our results showed that the addition of 0.006% Ca^2+^ at different times could promote the growth of the L2 strain, and there was no significant difference between each addition time (Figure 3c). With the increase in addition time, the production of erucamide also had a significant promoting effect (*p* < 0.05), which was first increased and then stabilized. Among them, at 6 h, the erucamide content was the highest, reaching 1.665 ± 0.017 mg/L, which was 25.57% higher than that of the control. Therefore, 6 h was selected as the most suitable adding time. The effect of adding 0.5% colza oil at different fermentation times on the production of extracellular erucamide is shown in Figure 3d. The results showed that the addition of 0.5% colza oil at different times significantly promoted the growth of bacteria and the content of erucamide (*p* < 0.05), which showed a trend of increasing first and then decreasing. After adding 0.5% colza oil at 6 h, the content of erucamide reached the maximum of 1.707 ± 0.012 mg/L, which was 27.67% higher than that of the blank control. Therefore, 0.5% colza oil was added 6 h after fermentation.

#### 3.1.4. Orthogonal Test to Optimize the Combination of Accelerant Addition

According to the results of the single factor analysis of various accelerants, Tween 80 (A), chloroform (B), colza oil (C), and Ca^2+^ (D) were selected as four factors, and three levels were investigated. The L_9_ (3^4^) orthogonal experiment was designed with the content of extracellular erucamide as the index. The orthogonal experiment design and results are shown in Table 4. The results of the range analysis showed that the main factor affecting the production of erucamide by the L2 strain was the concentration of Tween 80, followed by the concentration of Ca^2+^ and colza oil, and the minimum was the concentration of chloroform (Table 4). Through the orthogonal experiment, the optimal control conditions for erucamide production by the L2 strain were 0.3% Tween 80, 0.004% Ca^2+^, 0.04% colza oil, and 0.02% chloroform. Tween 80 and chloroform were added before fermentation, and Ca^2+^ and colza oil were added at 6 h of fermentation. The verification experiment was carried out at the level of regulatory combinations. Three repeated experiments were carried out with the blank group without any additives as the blank group. The yield of erucamide in the blank group was 1.341 mg/L. The experimental group reached 1.778 mg/L, which was 32.59% higher than the blank group and 60.26% higher than that before fermentation optimization, indicating that the regulatory conditions determined by the orthogonal experiment were reliable.

### 3.2. Study on the Mechanism of Accelerants in Cell Metabolism

#### 3.2.1. Effect on the Conductivity of Cell Suspension

Figure 4 shows the effect of the optimized accelerant combination on the conductivity of the cell suspension. The results showed that the conductivity of the bacterial suspension was significantly higher than that of the control group, and the treatment group and the blank control group showed an increasing trend. Specifically, there was no significant difference in the conductivity of the bacterial suspension between the treatment group and the blank group at 2 h, probably because the strain needed to adapt to the new environment when it was just inoculated into the fermentation medium. The conductivity after 8 h of treatment was significantly higher than that of the blank group, indicating that the permeability of the cell membrane of the L2 strain was changed after treatment with the accelerant, resulting in the leakage of intracellular electrolytes into the culture medium, which in turn increased the conductivity of the culture medium.

#### 3.2.2. Effects on Nucleic Acid and Protein Leakage

Changes in cell membrane permeability lead to a large amount of intracellular proteins leaking out of the cell. Therefore, measuring protein leakage becomes an important indicator of cell membrane permeability. It can be seen in Figure 5 that, within 0–12 h, the absorbance values of nucleic acids and proteins increased gradually with the increase in treatment time, and were significantly higher than those of the control group (Figure 5a,b). At 12 h, the nucleic acid and protein contents of the experimental group increased by 0.793 and 1.142 times, respectively, compared with the control group. The results showed that the addition of accelerants could change the cell membrane permeability of the L2 strain, resulting in the leakage of intracellular nucleic acids and proteins.

#### 3.2.3. Effect on the Permeability of Cell Membrane

Under normal circumstances, β-galactosidase, lactate dehydrogenase, alkaline phosphatase, and other active substances in the membrane will not be secreted to the outside of the cell. When the cell membrane changes, some intracellular enzymes will leak out of the cell. Therefore, by measuring the activity of β-galactosidase in the supernatant of the bacterial liquid, the infiltration of intracellular macromolecules into the outside of the cell is detected to illustrate the damage of the cell membrane. Figure 6 shows the effect of accelerants on the permeability of the cell membrane of the L2 strain. The results showed that the enzyme activity of β-galactosidase in the supernatant was higher than that in the control group after the treatment of the accelerant, and the enzyme activity of β-galactosidase was also enhanced with the prolongation of the treatment time, indicating that the addition of the accelerant had a certain destructive effect on the inner membrane of the cell, which changed its permeability and eventually led to the leakage of β-galactosidase to the outside of the cell.

#### 3.2.4. Effect on Fatty Acid Composition of Cell Membrane

The fluidity and integrity of the cell membrane are mainly determined by the fatty acyl length and saturation of the cell membrane fatty acids. The effect of the best combination of accelerants on the fatty acid composition of the cell membrane of the L2 strain was further studied. The fatty acid composition of the cell membrane of the L2 strain is shown in Table 5, with the fermentation medium without any accelerant as the control. It can be seen from the table that 9 fatty acids were detected in the control group, while 14 fatty acids were detected in the experimental group. Four unsaturated fatty acids (linoleic acid, r-linolenic acid, arachidonic acid, and linolenic acid) and one saturated fatty acid (tricosanoic acid) were detected in the control group. Among them, linoleic acid can balance the intestinal microflora and link the intestinal flora [30]. Tricosanoic acid has anticancer activity. It has been reported in plant oils such as crape myrtle seed oil and pomegranate seed oil [31,32]. In addition, after treatment with the accelerant, the content of different components of membrane fatty acids varied greatly. Compared with the control group, the content of saturated fatty acids in the experimental group increased from 0.719% of the control to 2.441%, and the content of saturated fatty acids decreased. In addition, the content of common unsaturated fatty acids increased from 0.848% of the control to 22.648%, and the content of unsaturated fatty acids decreased. The ratio of unsaturated fatty acid components to saturated fatty acid components in membrane lipids increased from 0.893 of the control to 1.856.

## 4. Discussion

The extracellular erucamide content was used as an index to study the effects of various accelerants on the production of erucamide by *B. megaterium* L2. A single accelerator can promote an increase in the production of metabolites [33,34]; however, we found that using accelerators in combination can significantly increase the production of metabolites. In this study, Tween 80, chloroform, colza oil, and Ca^2+^ concentrations were selected as four factors, and three levels were investigated. The results showed that the order of the primary and secondary factors affecting the production of erucamide was Tween 80 > Ca^2+^ > colza oil > chloroform. The optimal regulation conditions for the production of erucamide by *B. megaterium* L2 were 0.3% Tween 80, 0.004% Ca^2+^, 0.04% colza oil, and 0.02% chloroform. Under this condition, the content of erucamide reached 1.778 mg/L, which was 32.59% higher than that of the blank group and 60.26% higher than that before fermentation optimization.

A single accelerant may enhance metabolite production through various mechanisms. For instance, Tween 80, an ester of water-soluble long-chain fatty acids, acts as an accelerant by altering the arrangement and dispersion of the phospholipid bilayer in the cell membrane. This modification increases the fluidity of the membrane and reduces the resistance to the transport of nutrients and metabolites into the cell [35], thereby facilitating the production of erucamide. Similarly, chloroform can affect the cell membrane, leading to increased permeability for protons and ions. When proton power is insufficient, energy transmission is hindered, resulting in a compromised ability of the cell membrane to selectively exchange and regulate substances. This disruption impacts the membrane’s barrier function and energy conversion, causing erucamide to be released extracellularly [36]. Additionally, metal ions such as Ca^2+^ are vital for cell growth and metabolic synthesis. Ca^2+^ enhances the exchange of materials and energy across the membrane and increases the ionic strength of the culture medium, which facilitates nutrient uptake and target product release [37]. Furthermore, lipids found in vegetable oils can provide essential precursor substances for target metabolites through various enzymatic reactions, thereby promoting their increased production [38]. Notably, colza oil contains low levels of erucic acid, which serves as a precursor for erucamide synthesis within a specific concentration range.

Subsequently, the mechanism of erucamide production by the L2 strain was further investigated by measuring conductivity, nucleic acid and protein leakage, and β-galactosidase activity. The data showed that the conductivity of the bacterial solution was significantly increased, and the extracellular nucleic acid, protein, and β-galactosidase activities were higher than those of the control. These results indicated that the accelerants changed the permeability and selectivity of the cell membrane of the L2 strain, and the intracellular substances were released from the cells but did not cause damage or death to the bacteria. We propose that the observed increase in erucamide production may be attributed to the synergistic application of accelerants. Specifically, on the one hand, Tween 80 and chloroform alter the structure of the cell membrane of *B. megaterium* L2, enhancing both the fluidity and permeability of the membrane, which facilitates the release of nutrients and erucamide. On the other hand, Ca^2+^ and colza oil promote the synthesis and secretion of erucamide by improving the exchange of energy substances across the membrane, thereby supplying essential nutrients and precursor substances. In summary, the results of this study indicate that *B. megaterium* L2 has the potential for industrial production of erucamide, which can provide a basis for the industrial production of microbial erucamide.

## 5. Conclusions

This study successfully optimized the fermentation conditions for *B. megaterium* L2 to enhance its ability to produce erucamide by adding various accelerants. Through single-factor and orthogonal experiments, the optimal conditions were identified as 0.3% Tween 80, 0.004% Ca^2+^, 0.04% colza oil, and 0.02% chloroform. Under these conditions, the erucamide content reached 1.778 mg/L, representing a 32.59% increase compared with the blank group and a 60.26% increase over the pre-fermentation culture. The study further investigated the membrane metabolism mechanism of the L2 strain, revealing that the accelerants modified the membrane permeability without causing damage or killing the bacteria. Additionally, GC-MS analysis showed an increase in the ratio of unsaturated fatty acids to saturated fatty acids in the cell membrane, leading to enhanced membrane fluidity and reduced rigidity, which contributed to improved microbial metabolic activity and higher erucamide production.

## Figures and Tables

**Figure 1 microorganisms-13-00108-f001:**
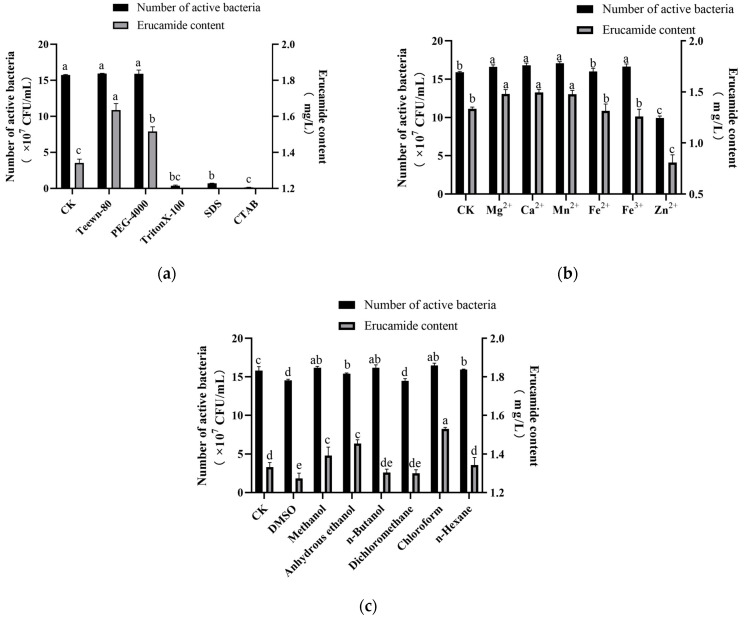
Effect of accelerants on erucamide fermentation. (**a**) Different kinds of surfactant accelerants; (**b**) different kinds of metal ion accelerants; (**c**) different kinds of organic solvent accelerants. Different lowercase letters indicate significant differences (*p* < 0.05).

**Figure 2 microorganisms-13-00108-f002:**
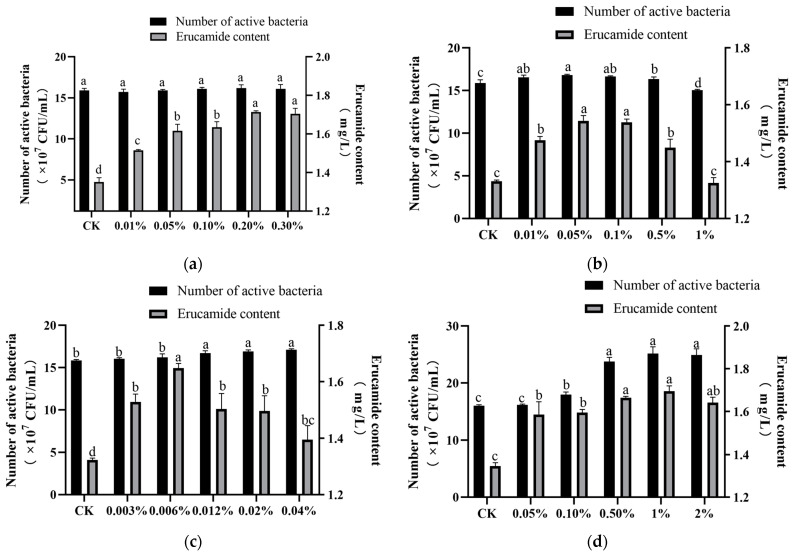
Effect of accelerant concentrations on erucamide fermentation. (**a**) Effect of Tween 80 addition amount on erucamide fermentation; (**b**) Effect of chloroform addition amount on erucamide fermentation; (**c**) Effect of Ca^2+^ addition amount on erucamide fermentation; (**d**) Effect of colza oil addition amount on erucamide fermentation. Different lowercase letters indicate significant differences (*p* < 0.05).

**Figure 3 microorganisms-13-00108-f003:**
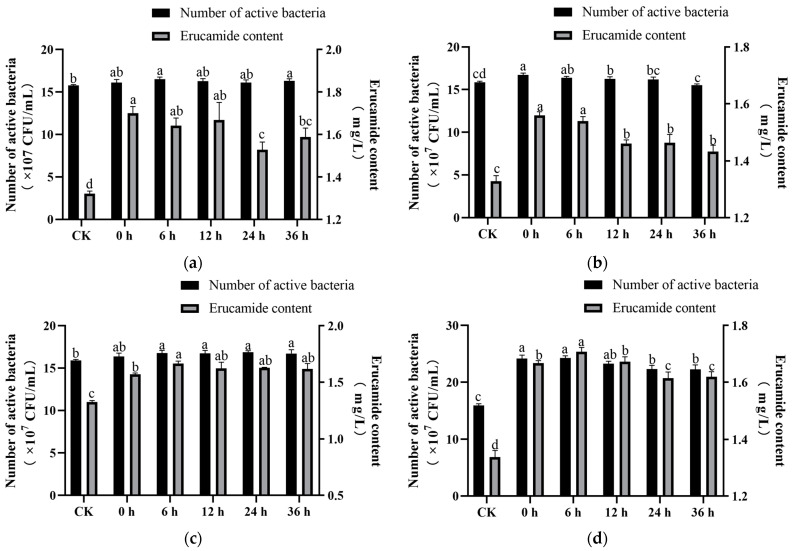
Effect of accelerant addition times on erucamide fermentation. (**a**) Effect of Tween 80 addition time on erucamide production; (**b**) Effect of chloroform addition time on erucamide production; (**c**) Effect of Ca^2+^ addition time on erucamide production; (**d**) Effect of Ca^2+^ addition time on erucamide production. Different lowercase letters indicate significant differences (*p* < 0.05).

**Figure 4 microorganisms-13-00108-f004:**
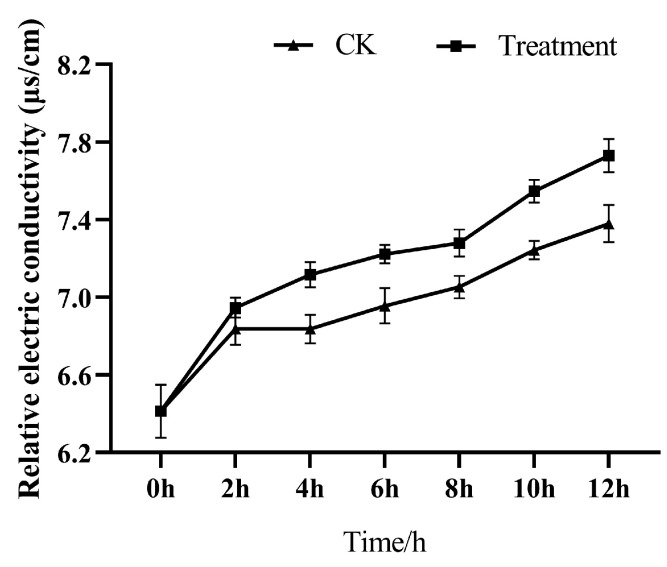
Effect of accelerants on the conductivity of L2 strain.

**Figure 5 microorganisms-13-00108-f005:**
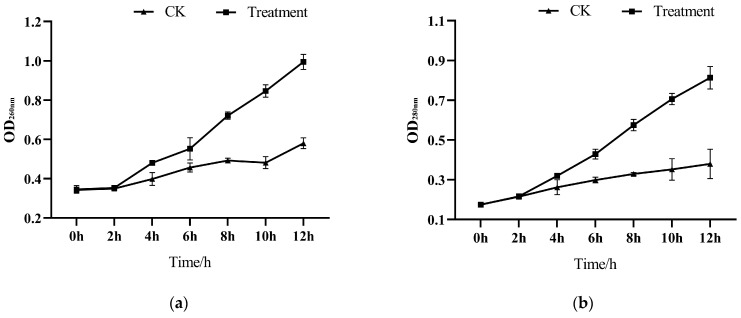
Effect of accelerants on (**a**) nucleic acid and (**b**) extracellular protein leakage of L2 strain.

**Figure 6 microorganisms-13-00108-f006:**
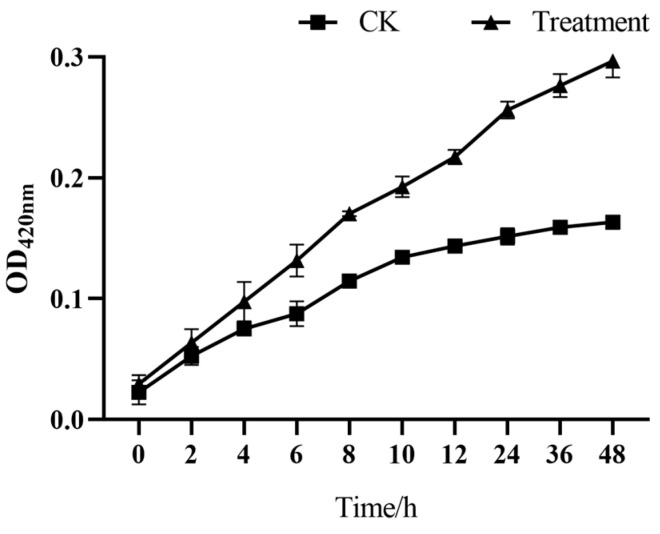
The penetrant effect of the accelerants on the cellular inner membrane.

**Table 1 microorganisms-13-00108-t001:** Accelerant types.

Accelerant Type	Accelerant Name
0.1% surfactant (g/L)	Tween 80, CTAB, SDS, PEG-6000, TritonX-100
0.1% organic solvent (*v*/*v*)	n-butanol, methanol, anhydrous, ethanol, chloroform, dichloromethane, n-hexane, DMSO
0.02% metal ions (g/L)	Mg^2+^, Ca^2+^, Mn^2+^, Fe^2+^, Fe^3+^, Zn^2+^
Colza oil	Set the addition amount to 0.05%, 0.1%, 0.5%, 1%, 2%

**Table 2 microorganisms-13-00108-t002:** Accelerants and their added amounts.

Accelerant Type	Accelerant Concentration
Tween 80 (g/L)	0.01%, 0.05%, 0.1%, 0.2%, 0.3%
Chloroform (*v*/*v*)	0.01%, 0.05%, 0.1%, 0.5%, 1%
Ca^2+^ (g/L)	0.003%, 0.006%, 0.012%, 0.02%, 0.04%
Colza oil (*v*/*v*)	0.05%, 0.1%, 0.5%, 1%, 2%

**Table 3 microorganisms-13-00108-t003:** Orthogonal factor level table L_9_(3^4^).

Level	Factor			
	A: Tween 80 (%)	B: Chloroform (%)	C: Colza Oil (%)	D: Ca^2+^ (%)
1	0.1	0.02	0.4	0.004
2	0.2	0.05	0.5	0.006
3	0.3	0.08	0.6	0.008

**Table 4 microorganisms-13-00108-t004:** Results and analysis of the orthogonal tests.

Test Number	A: Tween 80	B: Chloroform	C: Colza Oil	D: Ca^2+^	Content (mg/L)
**1**	2	2	2	1	1.758
**2**	3	3	3	1	1.600
**3**	1	3	2	2	1.722
**4**	2	1	3	2	1.775
**5**	3	1	2	3	1.584
**6**	3	2	1	2	1.670
**7**	2	3	1	3	1.670
**8**	1	2	3	3	1.631
**9**	1	1	1	1	1.535
**K1**	4.888	4.893	4.874	4.893	
**K2**	5.203	5.058	5.063	5.167	
**K3**	4.853	4.993	5.006	4.884	
$\overset{-}{K}$ **1**	1.629	1.631	1.625	1.631	
$\overset{-}{K}$ **2**	1.734	1.686	1.688	1.722	
$\overset{-}{K}$ **3**	1.618	1.664	1.669	1.628	
**Range**	0.117	0.033	0.063	0.091	
**Factors affecting**	A > D > C > B				
**Optimal conditions**	A_3_D_1_C_1_B_1_				1.778

**Table 5 microorganisms-13-00108-t005:** Analysis of fatty acid composition of cell membrane lipids of L2 strain.

Fatty Acid Type		Retention Time (min)	Fatty Acid Content (%)	
			Control Group	Test Group
Tetradecanoic acid	C14:0	36.223	1.134	0.857
9-Tetradecenic acid	C14:1	37.806	17.412	8.463
Pentadecanoic acid	C15:0	38.333	10.097	6.032
Palmitic acid	C16:0	40.895	17.123	12.990
Palmitoleic acid	C16:1	42.620	5.513	2.062
Stearic acid	C18:0	45.948	8.479	2.728
Oleic acid	C18:1	47.501	0.848	22.648
Linoleic acid	C18:2	49.818	-	9.493
r-Linolenic acid	C18:3n6	51.757	-	0.344
cis-11-Eicosenoic acid	C20:1	52.326	-	0.228
Linolenic acid	C18:3	52.632	-	4.132
Tricosanoic acid	C23:0	58.390	-	1.802
Tetracosanoic acid	C24:0	60.665	0.719	2.441
Docosahexaenoic acid	C22:6n3	69.975	9.755	2.547
Unsaturated/saturated fatty acids			0.893	1.856

Note: The number after C in the table represents the number of carbon elements in the fatty acyl group, and the number after the colon represents unsaturated fatty acid. “-” indicates that the component was not detected.

## Data Availability

All data generated or analyzed during this study are included in this published article.

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
