# Peer review of "Optimization of Fermentation Conditions for Increasing Erucamide Content in Bacillus megaterium Using Several Accelerants"

_microorganisms, 2025, doi:10.3390/microorganisms13010108_

Round 1
Reviewer 1 Report
Comments and Suggestions for Authors
Authors present intersting results related to optimization of erucamide content in Bacillus megaterium cultures, after application of different promoters. Results could be interesting to researchers in the field. Study is well planned and performed. Conclusions are supported by results. Figures are of good quality. Following comments should be addressed to improve the manuscript:
1. The media optimization is usually performed by surface response methodology (Design Expert package) or neural network algorithms. See as a samples following articles. Some information related to RSM and ANN-GA should be added to Introduction.
Hui-Lane Lau, Fadzlie Wong Faizal Wong, Raja Noor Zaliha Raja Abd Rahman, Mohd Shamzi Mohamed, Arbakariya B. Ariff, Siew-Ling Hii. Optimization of fermentation medium components by response surface methodology (RSM) and artificial neural network hybrid with genetic algorithm (ANN-GA) for lipase production by Burkholderia cenocepacia ST8 using used automotive engine oil as substrate. Biocatalysis and Agricultural Biotechnology. Volume 50, 2023, 102696, https://doi.org/10.1016/j.bcab.2023.102696.
Wang Z, Li N, Zhou X, Wei S, Zhu Y, Li M, Gong J, He Y, Dong X, Gao C, Cheng S. Optimization of fermentation parameters to improve the biosynthesis of selenium nanoparticles by Bacillus licheniformis F1 and its comprehensive application. BMC Microbiol. 2024 Jul 20;24(1):271. doi: 10.1186/s12866-024-03410-5.
Wu D, Fu L, Cao Y, Dong N, Li D. Genomic insights into antimicrobial potential and optimization of fermentation conditions of pig-derived Bacillus subtilis BS21. Front Microbiol. 2023 Sep 29;14:1239837. doi: 10.3389/fmicb.2023.1239837.
2. Why concentrations of colza oil in Table 1 and 2 are different?
3. Section 2.3.4 Provide citations for previous use of orthogonal method. Present also a method of calculation for this method. How values presented in Table 5 were calculated? Explain it.
4. Line 170, 180, 196- write Latin names of bacteria and other organisms in italics.
5. Line 193. Write Na2CO3.
6. Line 179- change the sentence: Draw a line chart, for example to: A line chart was drawn.
7. Table 5. Which factors are A, B, C, D? Describe it in the description of Table 5.
8. The Discussion section should be rewritten to show Authors own results in context of other researchers results, supported by appropriate citations. Show similarities and differences with other results, show reasons of it. Now it is rather a presentation of results, not their discussion.
Author Response
Comments 1: The media optimization is usually performed by surface response methodology (Design Expert package) or neural network algorithms. See as a samples following articles. Some information related to RSM and ANN-GA should be added to Introduction.
Response 1: Thank you for your correction. We have added the information about response surface methodology to the fermentation optimization section in the introduction and cited the literature recommended by the experts. (lines 96-97)
Comments 2: Why concentrations of colza oil in Table 1 and 2 are different?
Response 2: Thank you for your correction, this was a writing error on our part and has been corrected in Table 2.
Comments 3: Section 2.3.4 Provide citations for previous use of orthogonal method. Present also a method of calculation for this method. How values presented in Table 5 were calculated? Explain it.
Response 3: Thank you for pointing out our mistake. We have now numbered the accelerants in Section 2.3.4 and added a note below Table 5.
Comments 4: Line 170, 180, 196- write Latin names of bacteria and other organisms in italics.
Response 4: Thank you for pointing this out. We have completed the revision as suggested.
Comments 5: Line 193. Write Na2CO3.
Response 5: Thank you for your correction, we have completed the revision as suggested
Comments 6: Line 179- change the sentence: Draw a line chart, for example to: A line chart was drawn.
Response 6: Thank you for your correction, we comprehensively considered this comment of the three reviewers and deleted the sentence "with the treatment time as the abscissa and the absorbance as the ordinate. Plot the data as a line graph". (line 198)
Comments 7: Table 5. Which factors are A, B, C, D? Describe it in the description of Table 5.
Response 7: Thank you for your suggestion. We have changed the accelerators corresponding to the four values of A, B, C, and D in Table 5 to Tween 80, chloroform, colza oil, and Ca2+, and we have added the information in Table 5.
Comments 8: The Discussion section should be rewritten to show Authors own results in context of other researchers results, supported by appropriate citations. Show similarities and differences with other results, show reasons of it. Now it is rather a presentation of results, not their discussion.
Response 8: Thank you for your suggestion. We have written the conclusion and discussion separately and cited appropriate literature to support the conclusion.
Reviewer 2 Report
Comments and Suggestions for Authors
Please add detailed information about the aspects involving biosynthesis of the compound of interest by Bacillus megaterium.
Please add the importance of measuring b-galactosidase activity in the introduction section.
Please revise line 178-179
Add the importance of cell membrane of B. megaterium regarding erucamide biosynthesis
There is a misspelled word in methodology: Agligent. Check the whole document for small errors.
Please add the importance of protein ‘’leaking’’. Could it be water-soluble extracellular proteins produced by B. megaterium?
Please separate the discussion section according to methodology and results.
Please add references in the discussion section and explain the effects extensively and the references should be of very recent years.
Please add more references in the document of 2023-2024. There is only one 2023 reference and the rest is of diverse years of publication but relatively distant to recent times.
Please add concluding remarks
The article is interesting by its results and needs context on why several tests and assys were made and its impact on erucamide biosynthesis and detection.
Author Response
Comments 1: Please add detailed information about the aspects involving biosynthesis of the compound of interest by Bacillus megaterium.
Response 1: Thank you very much for your comments and suggestions. We have added the information as suggested, the added text reads as follows on (lines 91-95):
In previous studies, five bioactive compounds were isolated from B. megaterium L2 by the bioactivity-guided separation, and characterization of compounds was done by NMR (1H-NMR and 13C-NMR) and was identified as erucamide, behenic acid, palmitic acid, PAA and β-sitosterol and a method for the determination of erucamide was established[17,18].
Comments 2: Please add the importance of measuring β-galactosidase activity in the introduction section.
Response 2: Thank you for your suggestion, we have added the rationale and importance of measuring β-galactosidase at the beginning of Section 2.4.3 (lines 200-206).
Comments 3: Please revise line 178-179
Response 3: Thank you for your careful review. We have completed the revision in our manuscript as suggested
Comments 4: Add the importance of cell membrane of B. megaterium regarding erucamide biosynthesis
Response 4: Thank you for your suggestion. We have added a description of how changes in bacterial cell membranes affect metabolite production in the Introduction:
The cell membrane of microorganisms serves as a critical barrier, controlling the entry and exit of metabolites, ions, and other molecules. It is essential for maintaining cellular homeostasis, energy balance, and metabolism. Changes in membrane permeability can have profound effects on microbial metabolite production, as the movement of metabolites, substrates, and waste products is directly linked to membrane transport processes. Altering membrane permeability can influence the availability of precursors, the accumulation of metabolites, and the efficiency of microbial metabolism, which in turn affects industrial bioprocesses, antibiotic resistance, and cell survival.
Comments 5: There is a misspelled word in methodology: Agligent. Check the whole document for small errors.
Response 5: Thank you for pointing this out. We have completed the revision in Section 2.2 (line 121).
Comments 6: Please add the importance of protein “leaking”. Could it be water-soluble extracellular proteins produced by B. megaterium?
Response 6: Thank you for your suggestion, we have made the revisions to the manuscript as you requested. We have added the importance of protein leakage at the beginning of Section 3.2.2.
Comments 7: Please separate the discussion section according to methodology and results.
Response 7: Thank you for your valuable suggestions to improve the quality of our manuscript. We have completed the revision as suggestion.
Comments 8: Please add references in the discussion section and explain the effects extensively and the references should be of very recent years.
Response 8: Thank you for your suggestion. We have added several recent papers to the discussion section to support the conclusion.
Comments 9: Please add more references in the document of 2023-2024. There is only one 2023 reference and the rest is of diverse years of publication but relatively distant to recent times.
Response 9: Thank you for your suggestion. We have added several recent papers to the article.
Comments 10: Please add concluding remarks
Response 10: Thank you for your careful review. We have completed the revision in our manuscript as suggested.
Comments 11: The article is interesting by its results and needs context on why several tests and assys were made and its impact on erucamide biosynthesis and detection.
Response 11: Thank you for your careful review. We have completed the revision in our manuscript as suggested.
Reviewer 3 Report
Comments and Suggestions for Authors
This study investigates the effects of different agents (detergents, metal ions, organic solvents…) added to the fermentation medium to improve the efficiency of erucamide production by Bacillus megaterium L2 strain. As authors recognize, these studies are preliminarily. In my opinion, this study might become interesting, but the format and presentation of the manuscript is also preliminary. Substantial modifications would be introduced. The following comments would be considered in an eventual re-submission of this subject.
Major concerns
Line 18: Concentrations such as 1.778 mg/L and other yields are too accurate. Statistical analysis is missing. Same in figures such as 1.634 mg/L at line 231, 1.529 mg/L at line 249 and many other data in the different sections of results.
Lines 24-25: The unsaturated to saturated fatty acid ratio increased from 0.893 to 1.856. This Table contains a number of pitfalls and errors. Some of them:
There are two palmitic acids. One of them would be probably palmitoleic acid.
Octadeca acid would be stearic acid.
Arachidonic acid is not C20:1
These data are not definitive as 25-30% of the composition is unknown. The sum of both columns is 70-75%. Some important and probably occurring fatty acids such as arachidic, behenic, erucic or EPA acids are missing. Probably, they have not been detected or identified. In summary, a qualitative increase of the ration might be assumed, but quantitation is not warranted by the available data at Table 6.
Table 1: The format should be improved. Anhydrous is not a solvent. Dichloromethane should be one word. Delete "set the addition amount of...".
Table 2: The concentrations tested for Colza oil v/v are 0.05%, 0.1%, 0.5%, 1%, 5%. 2%, the apparent optimal one according to subsequent results is 0.5-2% depending on the different sections. 2% is a concentration never tested according to the Table. 5% is not shown at Figure 2D. This is not acceptable. Please, clarify.
Table 3 is unnecessary. These optimal conditions could be expressed in a new more informative Table 2. Tested addition times could be expressed in a Table legend (or added to lines 139-140).
Fig. 1: (A) Detergent concentration is not given. According to Table 1, different concentrations were tested. Same at Figure 1B, as Table 2 indicates different ion concentrations. Please, clarify.
Lines 252-253 and 255-256 are not consistent. They are opposite. Please, clarify.
Figure 3 shows that optimal erucamide concentration was achieved at time 0 for: (a) Tween 80 addition time; (b) chloroform addition time. So, the convenience of those agents (accelerators?) is doubtful and not consistent with previous results. This point should be clarified. In general, time is a confusing variable. I recommend the elimination of this set.
Table 5: Is rapeseed oil the same that colza oil?. Terminology would be uniform and clear. Data at the bottom of the Table 5 are confusing. At the column “test number”, there are number, after that Kn, and after that other Kn set with a upper hyphen. What is the meaning of those sets?. Probably, I do not clearly understand the orthogonal tests, but other future readers could have the same problem. Please, clarify and, if possible, simplify the conclusions. Until lines 419-420, it is difficult to know that the optimal regulation conditions for the production of erucamide by B. megaterium L2 were 0.3% Tween 80, 0.004% Ca2+, 0.04% colza oil and 0.02% chloroform. On the other hand, these optimal concentrations are not statiscally proven.
Figure 5: The use of 260 nm for (a) nucleic acid and 280 nm for (b) extracellular protein leakage is a oversimplification. Why 260 are total nucleic acid and 280 nm are extracellular protein leakage. Please, clarify.
Minor points
Legends of figures are poor. Some important details are missing
The word “promoters” is not the most convenient one for the agents used. Genetically, promoters have a different meaning, and its use in the abstract and throughout the text (some time accelerators) is confusing. Agents, effectors or whatever are more precise.
Some abbreviations should be defined. For instance, CK (is that the control or “blank group” according to the abstract terminology?) Log P (what p is?.
Line 179: Delete the sentence “Draw a line chart”. It is obvious.
Bacillus megaterium should be always written in italics letter.
Quality of Figure 1c should be improved.
Lines 203 and 205: The ration of methanol should be given for the basic and acid solutions used.
Author Response
Comments 1: Line 18: Concentrations such as 1.778 mg/L and other yields are too accurate. Statistical analysis is missing. Same in figures such as 1.634 mg/L at line 231, 1.529 mg/L at line 249 and many other data in the different sections of results.
Response 1: Thank you for pointing this out. We have completed the revision as suggested, and added the statistical analysis in our manuscript.
Comments 2: Lines 24-25: The unsaturated to saturated fatty acid ratio increased from 0.893 to 1.856. This Table contains a number of pitfalls and errors. Some of them:
There are two palmitic acids. One of them would be probably palmitoleic acid.
Octadeca acid would be stearic acid.
Arachidonic acid is not C20:1
These data are not definitive as 25-30% of the composition is unknown. The sum of both columns is 70-75% . Some important and probably occurring fatty acids such as arachidic, behenic, erucic or EPA acids are missing. Probably, they have not been detected or identified. In summary, a qualitative increase of the ration might be assumed, but quantitation is not warranted by the available data at Table 6.
Response 2: Thank you for your reply. During the writing process of our article, some very professional chemical terms were translated incorrectly due to translation reasons. Thank you for pointing them out. We have now corrected them as follows: the second palmitic acid was changed to palmitoleic acid, which is an unsaturated fatty acid; Octadeca acid was changed to Stearic acid; Arachidonic acid was replaced with cis-11-Eicosenoic acid. This table shows the completed experimental section, and we did not detect any of the substances you mentioned in the experiment.
Comments 3: Table 1: The format should be improved. Anhydrous is not a solvent. Dichloromethane should be one word. Delete "set the addition amount of...".
Response 3: Thank you for your reply. Also caused a misunderstanding during the article translation process, Anhydrous has been replaced with Ethanol absolute.
Comments 4: Table 2: The concentrations tested for Colza oil v/v are 0.05%, 0.1%, 0.5%, 1%, 5%. 2%, the apparent optimal one according to subsequent results is 0.5-2% depending on the different sections. 2% is a concentration never tested according to the Table. 5% is not shown at Figure 2D. This is not acceptable. Please, clarify.
Response 4: Thank you for your correction. This is a typo on our part. In the experimental design, we set the maximum amount of colza oil added to 2%. There is no experimental parameter of 5%. We are very sorry for causing you confusion.
Comments 5: Table 3 is unnecessary. These optimal conditions could be expressed in a new more informative Table 2. Tested addition times could be expressed in a Table legend (or added to lines 139-140).
Response 5: Thank you for your suggestion. We have deleted Table 3 and written the experimental design in text form in Section 2.3.3.
Comments 6: Fig. 1: (A) Detergent concentration is not given. According to Table 1, different concentrations were tested. Same at Figure 1B, as Table 2 indicates different ion concentrations. Please, clarify.
Response 6: Thank you for your advice. The results listed in Figure 1 are for screening different types of accelerators, and Figure 2 is for screening the amount of accelerator added. The two figures have different effects.
Comments 7: Lines 252-253 and 255-256 are not consistent. They are opposite. Please, clarify.
Response 7: Thank you for your correction. After careful review, we deleted the sentence "indicating that organic solvents had no positive correlation effect on the growth of L2 strain and the yield of erucamide.".
Comments 8: Figure 3 shows that optimal erucamide concentration was achieved at time 0 for: (a) Tween 80 addition time; (b) chloroform addition time. So, the convenience of those agents (accelerators?) is doubtful and not consistent with previous results. This point should be clarified. In general, time is a confusing variable. I recommend the elimination of this set.
Response 8: Thank you for your suggestion. At the beginning of the experimental design, we thought that setting up a test for the fermentation time could test the changes in the fermentation results after adding the accelerator. The results in Figure 3 show that the best fermentation effect was achieved at 0 hours after adding Tween 80, and the result was better than the blank treatment group without adding the accelerator. This result also tells us that the fermentation time after adding Tween 80 should not be too long.
Comments 9: Table 5: Is rapeseed oil the same that colza oil?. Terminology would be uniform and clear. Data at the bottom of the Table 5 are confusing. At the column “test number”, there are number, after that Kn, and after that other Kn set with a upper hyphen. What is the meaning of those sets?. Probably, I do not clearly understand the orthogonal tests, but other future readers could have the same problem. Please, clarify and, if possible, simplify the conclusions. Until lines 419-420, it is difficult to know that the optimal regulation conditions for the production of erucamide by B. megaterium L2 were 0.3% Tween 80, 0.004% Ca2+, 0.04% colza oil and 0.02% chloroform. On the other hand, these optimal concentrations are not statiscally proven.
Response 9: Thank you for pointing out our mistake. The terms rapeseed oil and colza oil should not appear in the manuscript. We have now changed them to colza oil. Rows K1, K2, and K3 in Table 5 represent the sum of indicators at each level of each factor. For example, K1 represents the sum of the values of test indicators corresponding to the "1" level. In the orthogonal test, calculate the sum of the experimental results at each level, that is, K1, K2, K3 on columns 1, 2, and 3, and find the range (R value) of K1, K2, K3, and K, and calculate The size of the R value is used to rank the order of factor significance, and the K value is compared to select the optimal level combination, that is, the best experimental conditions.
Comments 10: Figure 5: The use of 260 nm for (a) nucleic acid and 280 nm for (b) extracellular protein leakage is a oversimplification. Why 260 are total nucleic acid and 280 nm are extracellular protein leakage. Please, clarify.
Response 10: Thank you for your reply. The 260nm wavelength corresponds to the absorption of nucleic acids, while the 280nm wavelength corresponds to the absorption of proteins, mainly because different substances have different absorption characteristics under ultraviolet light.
The absorption characteristics of nucleic acids and proteins under ultraviolet light are mainly determined by their molecular structure. Nucleic acids (such as DNA and RNA) have the largest absorption peak at a wavelength of 260nm, because the bases in nucleic acid molecules (such as adenine, guanine, cytosine, and thymine) have strong absorption at a wavelength of 260nm. And proteins have the largest absorption peak at 280nm wavelength, because certain amino acids in protein molecules (such as tryptophan, tyrosine, and phenylalanine) have strong absorption at 280nm wavelength. The difference in absorption characteristics allows us to use a UV spectrophotometer to measure the concentrations of nucleic acids and proteins separately.
Comments 11: Legends of figures are poor. Some important details are missing
Response 11: Thank you for pointing this out. We have completed the revision as suggested.
Comments 12: The word “promoters” is not the most convenient one for the agents used. Genetically, promoters have a different meaning, and its use in the abstract and throughout the text (some time accelerators) is confusing. Agents, effectors or whatever are more precise.
Some abbreviations should be defined. For instance, CK (is that the control or “blank group” according to the abstract terminology?) Log P (what p is?.
Response 12: Thank you for your suggestion! Using the word "promoters" does have the risk of causing misunderstanding. We have replaced "promoters" with "accelerants", which can more clearly express the promoting effect of additives on the microbial fermentation process. The "CK" in the manuscript means blank group. We have marked the first appearance of the word "CK" in Section 3.1.1 of the manuscript to facilitate readers' understanding. LogP represents the distribution coefficient of organic solvents in n-octanol and water. As LogP has not described in the manuscript, we have now removed this value and redrawn Figure 1c. in the manuscript
Comments 13: Line 179: Delete the sentence “Draw a line chart”. It is obvious.
Response 13: Thank you for your suggestion, we have completed the revision as suggested (line 198).
Comments 14: Bacillus megaterium should be always written in italics letter.
Response 14: Thank you for your suggestions. There were omissions in our previous work, which have now been revised.
Comments 15: Quality of Figure 1c should be improved.
Response 15: Thank you for pointing this out. We have completed the revision as suggested.
Comments 16: Lines 203 and 205:The ration of methanol should be given for the basic and acid solutions used.
Response 16: Thank you for pointing this out. We have completed the revision as suggested (line 230, line 231).
Round 2
Reviewer 1 Report
Comments and Suggestions for Authors
Authors corrected the manuscript according to suggestions. Authors should check it all citations are correctly numbered in the text. For example Yu et al. cited in section 2.3.4 has number 21, but in the reference list it is 22.
Check numbers of all references; if numbers in citation list and text are appropriate.
Author Response
Comments 1: Authors corrected the manuscript according to suggestions. Authors should check it all citations are correctly numbered in the text. For example Yu et al. cited in section 2.3.4 has number 21, but in the reference list it is 22.
Check numbers of all references; if numbers in citation list and text are appropriate.
Response 1: Thank you for pointing out our mistake. We have corrected the incorrect reference number and checked the reference numbers throughout the paper.
Reviewer 2 Report
Comments and Suggestions for Authors
The information about b-galactosidase should be in the introductory section rather than the methodology. Other than that, the manuscript and cy correcting that aspect, the manuscript is ready for publishing
Author Response
Comments 1: The information about b-galactosidase should be in the introductory section rather than the methodology. Other than that, the manuscript and cy correcting that aspect, the manuscript is ready for publishing
Response 1: Thank you for your suggestion. We have moved the introduction of β-galactosidase from the Methodology section to the Introduction section (Lines 87-93).
Reviewer 3 Report
Comments and Suggestions for Authors
I appreciate the positive tone of the authors' response letter, which recognizes some typos and errors in the primitive version of the manuscript. Most of them have been corrected and I believe that the manuscript has improved in content and format. Some of the comments in the responses could still be further discussed, as perhaps authors did not understand the comment, but I think that in general the paper has been modified in a manner consistent with the comments and can be accepted. A couple of small bugs to modify before that:
Table 4: Replace Colaz by Colza
Conclusion, line 622- Use italics for B. megaterium and check all others possible lain names.
Thank you
Author Response
Comments 1: Table 4: Replace Colaz by Colza
Response 1: Thank you for pointing out our mistake! We have corrected Colaz in Table 4 to Colza
Comments 2: Conclusion, line 622- Use italics for B. megaterium and check all others possible lain names.
Response 2: Thank you for pointing out our mistake. We have italicized B. megaterium and reviewed the entire article.